# Evaluation of Paraspinal Muscle Degeneration on Pain Relief after Percutaneous Epidural Adhesiolysis in Patients with Degenerative Lumbar Spinal Disease

**DOI:** 10.3390/medicina59061118

**Published:** 2023-06-09

**Authors:** Misun Kang, Shin Hyung Kim, Minju Jo, Hyun Eom Jung, Jungbin Bae, Hee Jung Kim

**Affiliations:** Department of Anesthesiology and Pain Medicine, Anesthesia and Pain Research Institute, Yonsei University College of Medicine, Seoul 03722, Republic of Korea; kangmsane@yuhs.ac (M.K.); tessar@yuhs.ac (S.H.K.); jominju7@naver.com (M.J.); eom4531@yuhs.ac (H.E.J.); jungbinbae@yuhs.ac (J.B.)

**Keywords:** elderly patients, degeneration, epidural adhesiolysis, myosteatosis, sarcopenia, pain management

## Abstract

*Background and Objectives*: The analgesic effectiveness of epidural adhesiolysis may be influenced by morphological changes in the paraspinal muscles, particularly in elderly patients. The objective of this study was to assess whether the cross-sectional area or fatty infiltration of the paraspinal muscles impacts the treatment outcomes of epidural adhesiolysis. *Materials and Methods*: The analysis included a total of 183 patients with degenerative lumbar disease who underwent epidural adhesiolysis. Good analgesia was defined as a reduction in pain score of ≥30% at the 6-month follow up. We measured the cross-sectional area and fatty infiltration rate of the paraspinal muscles and divided the study population into age groups (≥65 years and <65 years). Variables were compared between the good and poor analgesia groups. *Results*: The results revealed that elderly patients experienced poorer analgesic outcomes as the rate of fatty infiltration in the paraspinal muscles increased (*p* = 0.029), predominantly in female patients. However, there was no correlation between the cross-sectional area and the analgesic outcome in patients younger than or older than 65 years (*p* = 0.397 and *p* = 0.349, respectively). Multivariable logistic regression analysis indicated that baseline pain scores < 7 (Odds Ratio (OR) = 4.039, 95% Confidence Interval (CI) = 1.594–10.233, *p* = 0.003), spondylolisthesis (OR = 4.074, 95% CI = 1.144–14.511, *p* = 0.030), and ≥ 50% fatty infiltration of the paraspinal muscles (OR = 6.576, 95% CI = 1.300–33.268, *p* = 0.023) were significantly associated with poor outcomes after adhesiolysis in elderly patients. *Conclusions*: Fatty degeneration of paraspinal muscles is correlated with inferior analgesic outcomes following epidural adhesiolysis in elderly patients, but not in young and middle-aged patients. The cross-sectional area of the paraspinal muscles is not associated with pain relief after the procedure.

## 1. Introduction

Lower back pain has been the leading cause of disability for the past 30 years, according to a systematic review [1]. In a community-based cohort study, approximately 36% of elderly patients reported lower back pain, and the frequency or intensity of the pain was associated with other musculoskeletal pain or comorbidities [2]. High socioeconomic costs are incurred in the management of lower back pain, and thus optimal treatment is essential.

Percutaneous epidural adhesiolysis could be an option to consider for patients who do not show improvement with conservative treatments. The original concept of this procedure included disruption of perineural fibrosis and inhibition of repeated scar formation on affected spinal levels [3]. Because this procedure has a higher cost burden than a single epidural steroid injection (ESI), most patients expect the analgesic benefit will last longer. Therefore, it is crucial to have clinical information about the factors linked to long-term pain relief following this procedure, as it aids in decision-making for both patients and clinicians.

Sarcopenia, a condition associated with aging, is characterized by a decline in muscle mass, strength, and/or physical performance. In the diagnosis of sarcopenia, changes in muscle quality are emphasized rather than muscle size [4]. Intramuscular fat infiltration, known as myosteatosis, has a negative correlation with muscle function and is associated with frailty in elderly patients [5]. The authors discovered that in elderly patients who underwent lumbar ESI, a higher level of fat infiltration in the paraspinal muscles was associated with a poorer analgesic response, while no such correlation was observed with the cross-sectional area (CSA) [6]. Based on this finding, we hypothesized that the CSA and fatty infiltration rate of the paraspinal muscles would have different effects on the outcome of lumbar epidural adhesiolysis, particularly in elderly patients.

Our study aimed to assess how paraspinal muscle degeneration affects the effectiveness of percutaneous lumbar epidural adhesiolysis in providing pain relief, taking into account the age of the patients. Additionally, we examined whether CSA or fat infiltration rate of the paraspinal muscles is a more useful predictor for analgesic outcome after treatment.

## 2. Materials and Methods

### 2.1. Study Population

Approval for this retrospective observational study was granted by the Institutional Review Board of Yonsei University Health System, Seoul, Republic of Korea (No. 4-2022-1494). The flow diagram of the study is shown in Figure 1. Patients who received percutaneous epidural adhesiolysis at our clinic from 2012 to 2021 were enrolled in the study. Adult patients who had received a diagnosis of degenerative lumbar spinal disease via MRI within 12 months preceding the procedure were included in the study. Patients who did not follow up within 6 months of the procedure or with incomplete electronic medical records were excluded. Additionally, patients with metal artifacts on Magnetic Resonance Imaging (MRI) at L3/4 level were excluded.

### 2.2. CSA and Fatty Infiltration Measures

The paraspinal muscles encompassing the multifidus and erector spinae muscles were quantitatively measured on T1-weighted axial MRI images as described in our previous work [6]. To measure the CSA, manual tracing was performed separately on the right and left paraspinal muscles of the axial MRI image at the level of the L3/4 disc (Figure 2A). The Image J program (Version 1.53n, NIH, Bethesda, MD, USA) was utilized to perform all measurements. On the same images, the percentage of paraspinal fatty infiltration was determined using the threshold technique with region of interest (ROI) tracing adapted from the previous study (Figure 2B) [7]. The degree of fatty infiltration was categorized into three grades: grade 1, indicating normal levels of intramuscular fat (0–10%), grade 2, indicating mild levels of intramuscular fat (10–50%), and grade 3, indicating severe levels of intramuscular fat (>50%) [8]. CSA and fatty infiltration rate were measured two times by two pain physicians (HJK and MJ) to improve the inter-observer reliability, and the mean values were recorded. Each observer was unaware of the results obtained by the other observer.

### 2.3. Fluoroscopy-Guided Lumbar Percutaneous Epidural Adhesiolysis

Preceding each epidural adhesiolysis, transforaminal epidurography was performed to identify filling defects corresponding to the symptomatic levels and abnormalities observed on MRI. Following the completion of sterile preparation and draping, a 22-gauge, 8 cm Quincke tip needle was carefully inserted and advanced beneath the pedicle, with intermittent fluoroscopic guidance in a tunnel view. An anteroposterior view was captured to ensure that the needle tip resided within the lateral half of the pedicle, and a lateral view was obtained to confirm proper placement of the needle tip in the anterior epidural space. To confirm the position, 1–2 mL of contrast media was administered at each level. Once the contrast was confirmed in the targeted area, real-time fluoroscopy was used to assess filling defects.

All adhesiolysis procedures were performed by two practitioners with at least 7 years of clinical experience following the standard procedure as previously described [9]. The needle insertion site was prepared with betadine and covered while the patient was lying in the prone position. Confirmation of the sacral hiatus was achieved by employing fluoroscopy in the lateral view. A 10-gauge guide Tuohy needle was advanced through the sacral hiatus into the sacral canal and the epidural space was ensured by injecting 3 mL of contrast media. After confirming, a steerable navigation catheter (ST. Reed plus^®^, Seawon meditech, Bucheon-si, Republic of Korea) was inserted through a guide needle. Upon reaching the target site, 3 mL of the contrast medium was injected to ensure that the catheter was well positioned in the epidural space and spread to the desired target site (Figure 3). After confirming correct epidurogram, a total of 10 mL of 1% lidocaine, a typical dose of 5 mg of dexamethasone and 1500 IU hyaluronidase mixture was slowly injected at each target site.

### 2.4. Patient Characteristics and Clinical Data Measurements

Patient demographics, pain-related data, and clinical data were gathered. Demographic data included age, sex, body mass index (BMI), documented medical comorbidities, and any history of lumbar surgery. Pain duration, baseline pain score using numeric rating scale (NRS), opioid usage for at least 1 month before adhesiolysis, presence of neurogenic intermittent claudication (NIC) or radicular pain, and good analgesic response on lumbar ESI were identified. The MRI results, evaluated by an independent radiologist, were analyzed for the presence of herniated lumbar disc, and central or foraminal stenosis with grading [10,11] and spondylolisthesis were analyzed. Additionally, patients who received lumbar surgery indicating a transition from conservative treatment to surgical treatment within 12 months of adhesiolysis were examined. In this study, a good analgesic efficacy after adhesiolysis was defined as a 30% or more decrease in pain score, which was regarded as a clinically significant improvement in pain intensity according to the NRS [12] at 6 months after adhesiolysis without an increase in analgesic medication.

### 2.5. Statistical Analysis

Descriptive data were expressed as mean ± standard deviation (SD) for continuous variables and as numbers (percentage) for categorical variables. Ordinal data and continuous variables that did not follow a normal distribution are presented as median and interquartile range (IQR). The normality of distribution was evaluated using the Shapiro–Wilk test. Patient characteristics, clinical data, and quantitative measurements of paraspinal muscle were analyzed using independent *t*-test, chi-squared test, or Fisher’s exact test when appropriate. The Mann–Whitney U test was used for continuous variables with non-normal distribution. A multivariable logistic regression model, with backward elimination, was constructed to identify the factors associated with analgesic outcome after adhesiolysis, and the adjusted odds ratio (aOR) and 95% confidence interval (CI) were calculated. All statistical analyses were conducted using the Statistical Package for the Social Sciences, version 26.0 (IBM Corp, Armonk, NY, USA). A value of *p* below 0.05 was considered statistically significant.

## 3. Results

From 2012 to 2021, 228 patients underwent fluoroscopy-guided lumbar epidural adhesiolysis at our clinic. Out of these, 28 patients were excluded due to meeting the exclusion criteria, resulting in a final enrollment of 200 patients. In 17 patients, analysis of MRI images was difficult due to artifacts. The remaining 183 patients were included in the study; 74 showed good analgesic outcomes and 109 showed poor analgesic outcomes after adhesiolysis (Figure 1).

Patients were categorized into two groups based on age: those aged below 65 years and those aged 65 years and above. Patient characteristics, clinical data, and quantitative measurements of the paraspinal muscles were compared between the group with good analgesia and the group with poor analgesia according to age (Table 1). In patients under the age of 65, poor analgesic outcome was observed when the baseline pain score was 7 or less. The patient’s age, sex, BMI, underlying disease including spinal surgery history, opioid usage, presence of NIC or radiating pain, efficacy in previous ESI, and specific findings on MRI did not differ between the good and poor analgesia groups. In patients aged 65 or older, poorer treatment outcome was similarly observed in patients with pain scores less than 7. Age, sex, BMI, underlying diseases including spinal surgical history, pain duration, opioid use, presence of NIC or radiating pain, previous ESI effects, and MRI findings were similar for both groups. The measurements of the paraspinal muscle CSA were comparable between the good analgesia group and the poor analgesia group regardless of patient age. However, higher fat infiltration rate of the paraspinal muscles in patients over the age of 65 was observed in patients with poor analgesic outcome compared to those with good analgesia; there were no significant differences in pain outcome based on the fat infiltration rate of the paraspinal muscles in the patient group under the age of 65.

We next examined whether the paraspinal muscle mass or fatty infiltration affected the analgesic outcome in patients aged 65 or older according to sex. A high fat infiltration rate of the paraspinal muscles was more frequently observed in female patients with poor analgesia after adhesiolysis, but not in male patients. No significant difference was observed in CSA between male and female patients. In patients under the age of 65, no significant difference in muscle mass or fat infiltration between analgesic efficacies was observed regardless of sex (Table 2).

Multivariable logistic regression analysis showed that baseline pain scores of <7 on NRS (aOR = 4.039, 95% CI = 1.594–10.233, *p* = 0.003), spondylolisthesis (aOR = 4.074, 95% CI = 1.144–14.511, *p* = 0.030), and more than 50% fat infiltration in the paraspinal muscle (aOR = 6.576, 95% CI = 1.300–33.268, *p* = 0.023) were independent factors associated with poor post-procedural outcomes of elderly patients. No significant factors related to the efficacy of the procedure was found in patients under the age of 65 (Table 3).

## 4. Discussion

The aim of our study was to identify the relationship between paraspinal muscle degeneration and pain relief following adhesiolysis in patients with degenerative lumbar disease. In this study, higher fatty infiltration of the paraspinal muscles appeared to be an independent factor associated with the analgesic efficacy of adhesiolysis in elderly patients. However, no association was observed between CSA of the paraspinal muscles and pain relief after the procedure.

Several factors related to the pain-relieving efficacy of adhesiolysis have been identified [3,13], but the role of spinal myosteatosis as a predictive factor in poor analgesic outcome was unclear. In a study conducted on geriatric hospitalized patients, the results showed that intramuscular fat infiltration has a negative association with muscle strength and function, indicating that fat infiltration may play an important role in sarcopenia [14]. Metabolic dysfunction induced by myosteatosis appears to be associated with chronic systemic inflammation conditions [15]. Patients with severely impaired chronic pain often exhibit a chronic low-grade inflammation condition [16]. Additionally, increased fat infiltration in the paraspinal muscles was linked to elevated expression of pro-inflammatory cytokines [17]. Therefore, both systemic and localized myosteatosis with inflammatory dysregulation might potentially affect the outcomes of the current study.

In this study, in patients aged 65 or older, a higher proportion of fat in the muscle was associated with a poor analgesic outcome; the CSA of the muscle was not related to the outcome. A recent study showed that the CSA of lumbar paraspinal muscles was not a significant predictor of disability or pain scores at 6- and 12-month follow-up in patients suffering from spinal stenosis [18]. In patients with lumbar disc disease, preoperative CSA did not affect surgical outcome, but a significant correlation was found between fatty degeneration and functional and subjective general conditions [19]. The results confirmed that the quality, not the quantity, of muscle is involved in a pathological mechanism that contributes to the occurrence of pain.

In the present study, we found that female patients with high fat infiltration rate had poor pain relief after adhesiolysis. Several studies have reported that women have higher lumbar paravertebral muscle fatty infiltration compared to men regardless of pain [20,21]. Furthermore, menopausal women after the age of 50 have a clear increase in the prevalence of myosteatosis [22]. To date, limited knowledge exists regarding the effect of sex on myosteatosis and response to pain procedures, but our results suggest that the relation of sex and myosteatosis have a definite effect on the treatment outcome.

However, paraspinal muscle CSA and quality did not show a significant correlation with the treatment outcome in patients under the age of 65. In a study of 650 Korean patients, the mean densities of multifidus muscle and erector spinae muscle were higher in the young age group compared with the old age group [23]. When the multivariate adaptive regression splines to the data, the decline rates in erector spinae density differed according to patient age (younger or older than 53 years) [24]. These findings suggest that age-dependent fat degeneration is significantly related to response to pain interventions, and further evaluation is needed.

Interestingly, in this study, geriatric patients with higher baseline NRS showed better pain relief after the procedure. A previous study reported that a positive outcome could be expected if patients with failed back surgery syndromes or spinal stenosis had pain scores lower than 9 [25]. However, in the previous study, the average duration of pain was 7.5 years, while in this study, the median duration of pain was 6 months. This suggests that the period of pain is related to the pain scores, but more evidence is needed to support this possibility.

There are several limitations to this study, including its retrospective design conducted at a single center, small sample size, and relatively short follow-up period. Approximately 16.9% of patients who underwent spinal surgery were included in the study, and these individuals may have unique pain mechanisms and increased paraspinal muscle fat accumulation. However, the presence or absence of spinal surgery did not impact the procedure’s effectiveness in this study. Further research is needed to investigate potential differences in treatment outcomes between patients with and without spinal surgery. The evaluation in this study focused on the cross-section of paraspinal muscles at the L3/4 level, but it is unclear if this accurately represents the total muscle mass and composition. Lastly, although patients with fluoroscopic image data from the procedure were included, detailed contrast dispersion patterns were not analyzed in this study.

## 5. Conclusions

Our results suggest that fatty infiltration rate of the paraspinal muscles, rather than the CSA, was negatively correlated with analgesic outcome after adhesiolysis in elderly patients. In young and middle-aged patient groups, paraspinal muscle degeneration did not appear to be significantly related to analgesic outcome. These results highlight the potential significance of geriatric rehabilitation, including muscle-strengthening exercise, in patients diagnosed with degenerative lumbar spinal disease.

## Figures and Tables

**Figure 1 medicina-59-01118-f001:**
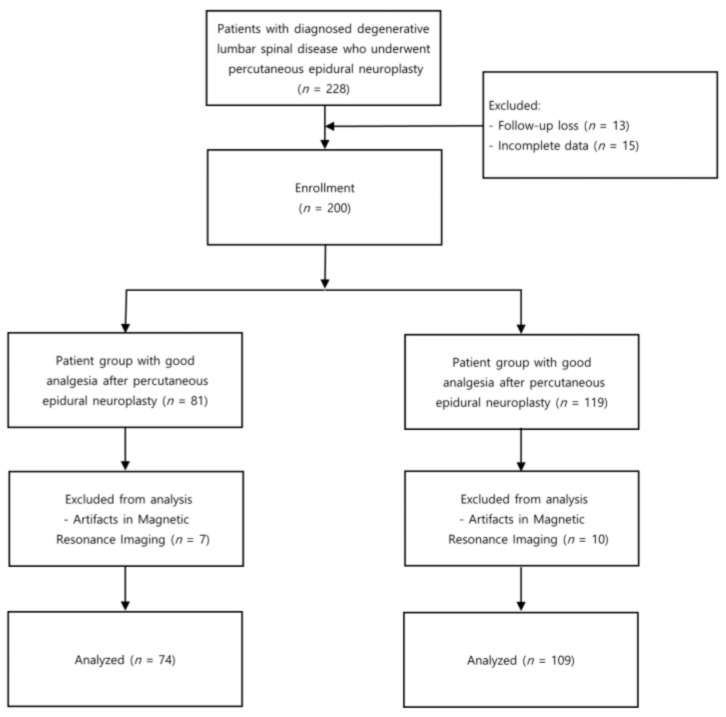
Flowchart of the study.

**Figure 2 medicina-59-01118-f002:**
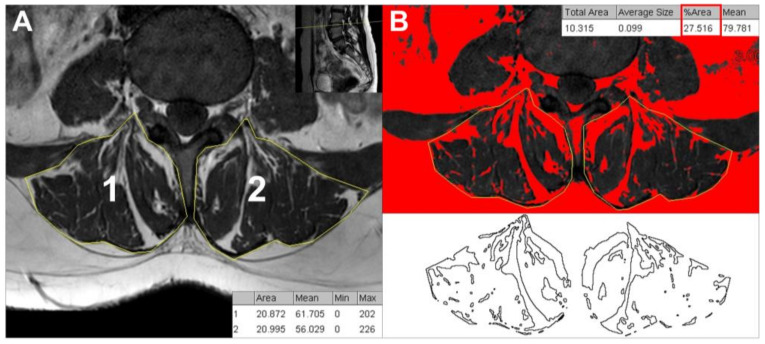
Quantitative measurements of cross-sectional area (**A**) and fatty infiltration rate (**B**) of the paraspinal muscles on a magnetic resonance image at the level of L3–L4 disc.

**Figure 3 medicina-59-01118-f003:**
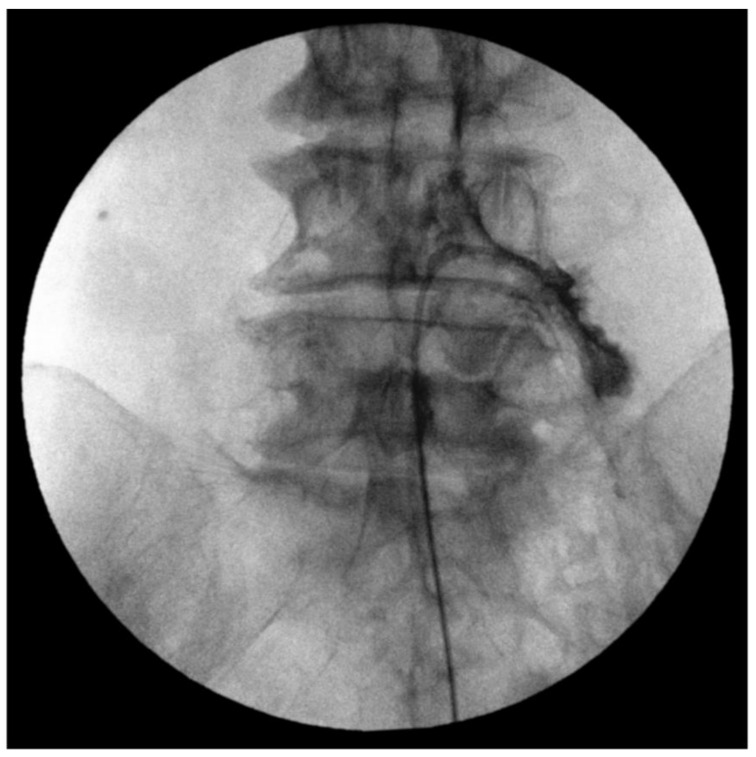
Epidurography pattern of percutaneous epidural adhesiolysis at the level of L4–L5 intervertebral foramen.

**Table 1 medicina-59-01118-t001:** Comparison of patient characteristics and clinical data between patients with good and poor analgesia after percutaneous epidural adhesiolysis according to patient age.

Variable	Age < 65 Group		*p*-Value	Age ≥ 65 Group		*p*-Value
	Good Analgesia (*n* = 36)	Poor Analgesia (*n* = 40)		Good Analgesia (*n* = 38)	Poor Analgesia (*n* = 69)	
**Patient characteristics**						
Age, years	53.69 ± 8.97 (33–64)	55.30 ± 9.26 (24–64)	0.446	75.29 ± 6.73 (65–91)	73.90 ± 5.68 (65–88)	0.259
Sex, M/F	21 (58.3%) /15 (41.7%)	18 (45.0%) /22 (55.0%)	0.246	16 (42.1%) /22 (57.9%)	28 (40.6%) /41 (59.4%)	0.878
Body mass index, kg/m^2^	24.30 (22.99;27.00)	24.88 (22.47;26.19)	0.999	25.15 (23.26;26.78)	24.21 (23.28;27.12)	0.728
<25	22 (61.1%)	21 (52.5%)	0.450	17 (44.7%)	42 (60.9%)	0.108
≥25	14 (38.9%)	19 (47.5%)		21 (55.3%)	27 (39.1%)	
Comorbid medical disease, *n*						
Cardiovascular disease	10 (27.8%)	15 (37.5%)	0.465	34 (89.5%)	60 (87.0%)	0.703
Diabetes mellitus	5 (13.9%)	5 (12.8%)	0.892	15 (39.5%)	20 (29.0%)	0.268
Osteopenia/osteoporosis	1 (2.8%)	2 (5.0%)	0.622	4 (10.5%)	5 (7.2%)	0.560
Spine surgery history, *n*	9 (25.0%)	6 (15.0%)	0.277	6 (15.8%)	10 (14.5%)	0.857
**Pain-related data**						
Pain duration, months	3.00 (1.00;10.75)	4.50 (2.00;12.00)	0.143	6.00 (2.00;12.00)	7.00 (3.00;30.00)	0.305
Baseline pain score, NRS 0–10	7.42 ± 1.68	6.48 ± 2.08	0.034	7.34 ± 1.76	5.70 ± 1.79	<0.001
NRS < 7	10 (27.8%)	20 (50.0%)	0.048	13 (34.2%)	46 (66.7%)	0.001
NRS ≥ 7	26 (72.2%)	20 (50.0%)		25 (65.8%)	23 (33.3%)	
Opioid usage, *n*	23 (63.9%)	29 (72.5%)	0.466	32 (84.2%)	51 (73.9%)	0.222
Presence of NIC, *n*	12 (33.3%)	13 (32.5%)	0.938	20 (52.6%)	44 (63.8%)	0.261
Presence of radiating pain, *n*	33 (91.7%)	38 (95.0%)	0.558	38 (100.0%)	63 (91.3%)	0.061
Good effect of previous ESI, *n*	16 (44.4%)	16 (40.0%)	0.817	18 (47.4%)	33 (47.8%)	0.964
**Pre-procedural MRI findings, *n***						
Herniated disc	31 (86.1%)	37 (92.5%)	0.365	37 (97.4%)	62 (89.9%)	0.157
Foraminal stenosis			0.999			0.264
None to mild	27 (75.0%)	30 (75.0%)		15 (39.5%)	35 (50.7%)	
Moderate to severe	9 (25.0%)	10 (25.0%)		23 (60.5%)	34 (49.3%)	
Central stenosis			0.594			0.096
None to mild	29 (80.6%)	30 (75.0%)		24 (63.2%)	32 (46.4%)	
Moderate to severe	7 (19.4%)	10 (25.0%)		14 (36.8%)	37 (53.6%)	
Spondylolisthesis	4 (11.1%)	4 (10.0%)	0.876	4 (10.5%)	18 (26.1%)	0.058
Transition to spine surgery within 1 year, *n*	6 (16.7%)	7 (17.9%)	0.884	8 (21.1%)	12 (17.6%)	0.667
**Paraspinal muscle CSA, cm^2^**						
Right	21.98 ± 4.87	20.81 ± 5.90	0.351	18.99 ± 3.44	19.82 ± 3.93	0.275
Left	21.87 ± 4.92	20.94 ± 6.00	0.461	18.43 ± 4.52	19.05 ± 4.00	0.462
Total	43.85 ± 9.55	41.74 ± 11.77	0.397	37.42 ± 7.53	38.87 ± 7.73	0.349
**Fatty infiltration%**						
Right			0.281			0.053
<10%	8 (22.2%)	7 (17.5%)		5 (13.2%)	15 (21.7%)	
10–50%	22 (61.1%)	20 (50.0%)		23 (60.5%)	25 (36.2%)	
≥50%	6 (16.7%)	13 (32.5%)		10 (26.3%)	29 (42.0%)	
Left			0.429			0.016
<10%	7 (19.4%)	13 (32.5%)		13 (34.2%)	11 (15.9%)	
10–50%	22 (61.1%)	21 (52.5%)		23 (60.5%)	46 (66.7%)	
≥50%	7 (19.4%)	6 (15.0%)		2 (5.3%)	12 (17.4%)	
Total			0.785			0.029
<10%	7 (19.4%)	10 (25.0%)		9 (23.7%)	8 (11.6%)	
10–50%	22 (61.1%)	24 (60.0%)		25 (65.8%)	43 (62.3%)	
≥50%	7 (19.4%)	6 (15.0%)		4 (10.5%)	18 (26.1%)	

Values are presented as mean ± SD, median (interquartile range) or number of patients (%). ESI, epidural steroid injection; NRS, numeric rating scale; NIC, neurogenic intermittent claudication; MRI, magnetic resonance imaging; CSA, cross-sectional area.

**Table 2 medicina-59-01118-t002:** Sex-specific comparison of fatty infiltration% and paraspinal muscle CSA between patients with good and poor analgesia after percutaneous epidural adhesiolysis according to patient age.

	Male			Female		
Age ≥ 65 group	Good analgesia (*n* = 16)	Poor analgesia (*n* = 28)	*p*-value	Good analgesia (*n* = 22)	Poor analgesia (*n* = 41)	*p*-value
Paraspinal muscle CSA, cm^2^	40.54 ± 6.70	42.89 ± 7.65	0.313	35.15 ± 7.41	36.13 ± 6.58	0.588
Fatty infiltration%			0.895			0.040
<10%	4 (25.0%)	5 (17.9%)		5 (22.7%)	3 (7.3%)	
10–50%	10 (62.5%)	19 (67.9%)		15 (68.2%)	24 (58.5%)	
≥50%	2 (12.5%)	4 (14.3%)		2 (9.1%)	14 (34.1%)	
Age < 65 group	Good analgesia (*n* = 21)	Poor analgesia (*n* = 18)	*p*-value	Good analgesia (*n* = 15)	Poor analgesia (*n* = 22)	*p*-value
Paraspinal muscle CSA, cm^2^	49.78 ± 12.22	48.71 ± 8.53	0.749	37.05 ± 6.25	35.16 ± 5.97	0.360
Fatty infiltration%			0.646			0.789
<10%	6 (28.6%)	7 (38.9%)		1 (6.7%)	3 (13.6%)	
10–50%	12 (57.1%)	10 (55.6%)		10 (66.7%)	14 (63.6%)	
≥50%	3 (14.3%)	1 (5.6%)		4 (26.7%)	5 (22.7%)	

Values are presented as mean ± SD, median (interquartile range) or number of patients (%). CSA, cross-sectional area.

**Table 3 medicina-59-01118-t003:** Factors associated with poor analgesia following lumbar epidural adhesiolysis in patients aged 65 years and above: result of multivariable logistic regression analysis.

	Adjusted Odds Ratio	95% Confidence Intervals	*p*-Value
Baseline pain score, <7 on NRS	4.039	1.594–10.233	0.003
Spondylolisthesis, yes	4.074	1.144–14.511	0.030
Total fatty infiltration%			
<10% (reference)	1.000		
≥50%	6.576	1.300–33.268	0.023

## Data Availability

Data are available upon request to the corresponding author.

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
