# Peer review of "Evaluation of Paraspinal Muscle Degeneration on Pain Relief after Percutaneous Epidural Adhesiolysis in Patients with Degenerative Lumbar Spinal Disease"

_medicina, 2023, doi:10.3390/medicina59061118_

Round 1

Reviewer 1 Report

Interesting and well conducted study. 

Authors forgot to delete the discussion instructions form in lines 229-232.

Did you consider to evaluate the sarcopaenia also basing on the PLVI parameter? It's a very useful parameter since it helps to redude the possible bias related to the differences in statural height and proportions among individuals, it has been already described in other Orthopaedic Surgery fields like hip replacements (doi: 10.1016/j.arth.2018.09.037). 

An additional area of interest is the risk of infections in sarcopenic patients. Is it possible to have an analysis regarding the risk of infection in the needle tract (although rare) or regarding surgical site infections in the patients that ultimately underwent to spine surgery or to other invasive procedures after the adhesiolisis? It may be possibile that the accidental introduction of pathogens with the adhesiolisis may generate asymptomatic bacteraemias in frail patients. It is a field that has seen increasing interest in the recent years, and particularly regarding the risk of surgical site infections in sarcopenic patients undergoing spine surgery there are contradditory reports (doi: 10.3928/01477447-20160811-02doi: 10.3171/2015.7.FOCUS15257; doi: 10.3390/microorganisms10101905)

Author Response

1. Authors forgot to delete the discussion instructions form in lines 229-232.

→ As for your comment, we have removed the redundant content.

2. Did you consider to evaluate the sarcopaenia also basing on the PLVI parameter? It's a very useful parameter since it helps to redude the possible bias related to the differences in statural height and proportions among individuals, it has been already described in other Orthopaedic Surgery fields like hip replacements (doi: 10.1016/j.arth.2018.09.037).

→ Thank you for your detailed review. We appreciate your suggestion regarding the measurement of psoas muscle cross-sectional area and the potential of using PLVI as an effective means to assess musculoskeletal muscle mass and identify sarcopenia. While we followed the paraspinal muscle mass measurement method recommended by other authors in our study (doi: 10.1097/00007632-200104150-00014; doi: 10.1186/s12891-019-2551-y), we acknowledge the benefits of PLVI to mitigate biases. We will take your advice into consideration for our future research endeavors.

3. An additional area of interest is the risk of infections in sarcopenic patients. Is it possible to have an analysis regarding the risk of infection in the needle tract (although rare) or regarding surgical site infections in the patients that ultimately underwent to spine surgery or to other invasive procedures after the adhesiolysis? It may be possible that the accidental introduction of pathogens with the adhesiolysis may generate asymptomatic bacteraemias in frail patients. It is a field that has seen increasing interest in the recent years, and particularly regarding the risk of surgical site infections in sarcopenic patients undergoing spine surgery there are contradictory reports (doi: 10.3928/01477447-20160811-02; doi: 10.3171/2015.7.FOCUS15257; doi: 10.3390/microorganisms10101905)

→ Thank you for the additional information. As you rightly pointed out, sarcopenia is an area that necessitates further research due to the uncertain findings regarding the risk of surgical site infection, despite its established association with prolonged hospitalization and increased mortality. Moreover, the presence of 33 patients who underwent surgery within a year following the adhesiolysis in this study highlights the potential for future research in related fields.

Reviewer 2 Report

This cross-sectional study examined whether fatty infiltration of the paraspinal muscles affects the results of epidural adhesiolysis. The manuscript is written in a clear manner. I have just minor comments for this manuscript.

Abstract: the sample size should be included in the abstract.

Methods: was the sample size calculated before the study starts?

Results: Line 173: do not start the sentence with a numerical (28).

For the multiple regression model, the 95% confidence intervals are very wide. I wonder if the authors have screened out multivariate outliers in the model. If not, please comment on the consistency of the data.

Discussion: The first paragraph of the discussion is from the template. Please delete.

I would like the authors to add on the possible biological relationship between myosteatosis and poor pain relief. 

Author Response

This cross-sectional study examined whether fatty infiltration of the paraspinal muscles affects the results of epidural adhesiolysis. The manuscript is written in a clear manner. I have just minor comments for this manuscript.

1. Abstract: the sample size should be included in the abstract.

→ We agree with your comment. Accordingly, we have included the sample size in the abstract.

-----

Materials and Methods: A total of 183 patients with degenerative lumbar disease who underwent epidural adhesiolysis were enrolled in the analysis.

2. Methods: was the sample size calculated before the study starts?

→ This study was conducted retrospectively, and the sample size was not determined in advance. All eligible patients who underwent the adhesiolysis from 2012 to 2021 were enrolled in the study, except for those who met the exclusion criteria.

3. Results: Line 173: do not start the sentence with a numerical (28).

→ We absolutely agree your opinion. We have modified the sentence.

------

Out of these, 28 patients were excluded due to meeting the exclusion criteria, resulting in a final enrollment of 200 patients.

4. For the multiple regression model, the 95% confidence intervals are very wide. I wonder if the authors have screened out multivariate outliers in the model. If not, please comment on the consistency of the data.

→ We had a thorough and meaningful discussion with the statistician regarding the topic you raised. Considering the nature of the three variables involved in the multivariate regression model, which happen to be categorical data, it seems unnecessary to delve into outlier discussion. Furthermore, upon examining the data distribution for each variable, it becomes apparent that there is not significant skewness present. Consequently, the consistency of the data remains intact without any concerns.

5. Discussion: The first paragraph of the discussion is from the template. Please delete.

→ As for your comment, we have removed the redundant content.

6. I would like the authors to add on the possible biological relationship between myosteatosis and poor pain relief.

→ Thank you for your comprehensive review. However, I would like to point out that the information regarding the biological relationship between myosteatosis and poor pain relief, which you recommended, is already incorporated in the text (line #241-251). Studies suggest that myosteatosis is associated with reduced muscle strength/function and chronic systemic inflammation, which can contribute to sarcopenia and impaired chronic pain. Furthermore, higher fat infiltration in the paraspinal muscles is linked to increased expression of pro-inflammatory cytokines. Consequently, both systemic and local myosteatosis with inflammatory dysregulation may affect the outcomes of the current study.
